# Position Classification of the Endotracheal Tube with Automatic Segmentation of the Trachea and the Tube on Plain Chest Radiography Using Deep Convolutional Neural Network

**DOI:** 10.3390/jpm12091363

**Published:** 2022-08-24

**Authors:** Heui Chul Jung, Changjin Kim, Jaehoon Oh, Tae Hyun Kim, Beomgyu Kim, Juncheol Lee, Jae Ho Chung, Hayoung Byun, Myeong Seong Yoon, Dong Keon Lee

**Affiliations:** 1School of Medicine, College of Medicine, Hanyang University, Seoul 04763, Korea; 2Department of Computer Science, Hanyang University, Seoul 04763, Korea; 3Department of Emergency Medicine, College of Medicine, Hanyang University, Seoul 04763, Korea; 4Machine Learning Research Center for Medical Data, Hanyang University, Seoul 04763, Korea; 5Department of Otolaryngology—Head and Neck Surgery, College of Medicine, Hanyang University, Seoul 04763, Korea; 6Department of HY-KIST Bio-Convergence, College of Medicine, Hanyang University, Seoul 04763, Korea; 7Department of Emergency Medicine, Seoul National University Bundang Hospital, Seongnam-si 13620, Korea

**Keywords:** intubation, endotracheal tube, deep learning, machine learning, artificial intelligence

## Abstract

Background: This study aimed to develop an algorithm for multilabel classification according to the distance from carina to endotracheal tube (ETT) tip (absence, shallow > 70 mm, 30 mm ≤ proper ≤ 70 mm, and deep position < 30 mm) with the application of automatic segmentation of the trachea and the ETT on chest radiographs using deep convolutional neural network (CNN). Methods: This study was a retrospective study using plain chest radiographs. We segmented the trachea and the ETT on images and labeled the classification of the ETT position. We proposed models for the classification of the ETT position using EfficientNet B0 with the application of automatic segmentation using Mask R-CNN and ResNet50. Primary outcomes were favorable performance for automatic segmentation and four-label classification through five-fold validation with segmented images and a test with non-segmented images. Results: Of 1985 images, 596 images were manually segmented and consisted of 298 absence, 97 shallow, 100 proper, and 101 deep images according to the ETT position. In five-fold validations with segmented images, Dice coefficients [mean (SD)] between segmented and predicted masks were 0.841 (0.063) for the trachea and 0.893 (0.078) for the ETT, and the accuracy for four-label classification was 0.945 (0.017). In the test for classification with 1389 non-segmented images, overall values were 0.922 for accuracy, 0.843 for precision, 0.843 for sensitivity, 0.922 for specificity, and 0.843 for F1-score. Conclusions: Automatic segmentation of the ETT and trachea images and classification of the ETT position using deep CNN with plain chest radiographs could achieve good performance and improve the physician’s performance in deciding the appropriateness of ETT depth.

## 1. Introduction

Endotracheal tube (ETT) intubation is an important procedure for airway management and ventilation of emergent and critical patients [1]. The distal tip of the ETT should not be placed in the esophagus, but in the trachea, and the correct depth position is approximately 30–70 mm from the carina, which would be in the middle third of the trachea in adult patients [2,3,4]. If the tip of the ETT is inserted too deeply, it could cause bronchial intubation, which leads to hyperinflation of the ipsilateral lung with atelectasis of the other, and barotrauma and hypoxemia could occur, resulting in an increased risk of mortality [5,6,7]. If the tip is inserted too shallowly in the first third of the trachea, it could cause accidental extubation, laryngeal spasm, and aspiration pneumonia, depending on changes in the patient’s head position [7,8,9].

Methods for ensuring correct positioning of the ETT include physical examination methods (auscultation of chest and epigastrium, visualization of thoracic movement, fogging in the tube), pulse oximetry, chest radiography, capnography, ultrasound, bronchoscopy, etc. [7,10,11,12,13,14]. Physical examination methods, as well as pulse oximetry and chest radiography, are not sufficiently reliable. Capnography is one of the best methods to confirm ETT placement in the trachea and can be used in patients who have adequate tissue perfusion [10,11,12,13,14,15]. Ultrasound is one of the portable and repeatable screening methods for the same purpose and may be used by someone who is experienced in this technique [14]. However, these methods cannot measure the distance from the carina to the tip of the ETT. Bronchoscopy is the gold standard for confirming the correct position of ETT and is always not available [15]. The chest radiograph is not the gold standard for this purpose, but certainly forms an important part of the overall assessment because it can be measured on an image [2,7]. However, measuring the distance on an image from chest radiography may be difficult for a novice clinician, or even for an expert, when working with poor image quality or under work overload conditions [16].

Recently, several studies reported on computer-aided assessment of ETT tube presence and position on chest radiography images using deep convolutional neural networks (CNN) [17,18,19,20]. They almost all showed that the network performed well in assessing ETT presence or absence with more than 0.97~0.99 accuracy, whereas its performance in assessing the proper positioning of the ETT tip was lacking or poorer. The technique of segmentation or localization of the carina and the ETT tip could determine malposition of the ETT or measure the distance between them and might improve performance in the classification of the ETT position [20]. There has been no research on the automatic segmentation of trachea and ETT images or on classification of proper positioning of the ETT tip. This study aims to develop an algorithm for the multi-label classification of ETT tip position (absence, shallow, proper, and deep position of ETT) with the application of automatic segmentation of the trachea and the ETT on chest radiographs using deep CNN.

## 2. Methods

### 2.1. Study Design

This is a retrospective study using plain chest anteroposterior (AP) radiography for segmentation of the trachea and the ETT tube and using CNN to determine proper positioning between them. The study was conducted at one regional emergency center of the tertiary hospital in Seoul (Seoul, Korea) between June and December 2021, and used data from January 2018 to March 2021. This study was approved by the Institutional Review Boards (IRB) of Hanyang University Hospital (ref. no. 2021-05-040) and the requirements for informed consent were waived by the IRBs of Hanyang University Hospital. All methods and procedures were carried out in accordance with the Declaration of Helsinki.

### 2.2. Dataset of Participants

A flowchart of data collection and analysis is shown in Figure 1.

Using the airway registry, we sorted and gathered chest AP images from a list of patients who underwent ETT intubation in the emergency room between January 2018 and March 2021 [21]. Images with the ETT were extracted from chest radiographs that had been used to check for confirmation or follow-up after intubation, and images without ETT were extracted from chest radiographs that had been taken for examination before intubation or after extubation of the ETT. We excluded images with tracheostomy tubes or images of poor quality due to severe damage to anatomical structures or massive subcutaneous emphysema by trauma. However, we did not exclude images with electrodes or nasogastric tubes or with central vein catheters. Two authors (Jung H.C., and Oh J.) reached consensus on each position of the carina and the distal tip of ETT and measured the distance between them in millimeters (mm) using the scale in the picture archiving and communication system (PACS, Centricity, GE Healthcare, Milwaukee, WI, USA). These images were categorized into four classes according to the ETT position, as follows: (1) absence (absence of the ETT on the image), (2) shallow position (more than 70 mm), (3) proper position (between 30 mm and 70 mm), (4) deep position (less than 30 mm). These images were extracted and stored as digital imaging and communication and medicine (DICOM)-format images using the PACS system. No personal information was included when saving images for data collection, and data were obtained without personal identifying data. In addition, arbitrary numbers were assigned to images, which were then coded and managed.

### 2.3. Segmentation of the Trachea and the ETT on Images and Data Processing

The author (Jung H.C.) performed a binary mask by painting the entire trachea area and ETT on images in consensus with another author (Oh J.) for the segmentation of the trachea and ETT. Both the trachea and ETT were segmented on images with ETT, whereas only the trachea was segmented on images without ETT, and these segmented images were stored as Nifty (Neuroimaging Informatics Technology Initiative) using the AVIEW program (AVIEW-Research, Corelinesoft, Seoul, Korea) (Figure 2). Moreover, we converted the segmented binary mask into COCO Dataset format [22]. To be specific, the binary mask was converted to a 2D polygon according to the COCO Dataset format. To obtain a 2D polygon, we computed the contour coordinates of the mask using the marching squares algorithm [23] and linearly interpolated them.

### 2.4. Proposed Models for Classification of Four Classes of ETT Position with Application of Automatic Segmentation Using Deep CNN

Classification of the ETT position on chest radiograph images is a typical multi-label task. Therefore, we used a classifier based on EfficientNet [24], which took only a single chest radiograph input image as our baseline. To further elevate the performance of the depth classification task, we proposed a 2-stage approach, which first segmented the trachea and the ETT areas, then used the segmentation results for the classification task. For the segmentation, we used the fully pretrained Mask R-CNN [25] as initial reference and fine-tuned it with our own dataset to segment out the trachea and the ETT regions. For the classification, the input image and binary masks of trachea and tube detected by our segmentation network were concatenated, and they were fed into the classification network (i.e., EfficientNet) based on inputs. The overall flow of the proposed model is illustrated in Figure 3.

#### 2.4.1. Network for the Segmentation of the Trachea and ETT

For the segmentation of the trachea and ETT areas, we employed conventional Mask R-CNN with the ResNet50 [26] feature pyramid network (FPN) [27] pretrained on the ImageNet dataset [28]. By using our own training dataset, we further updated the initial parameters of Mask R-CNN. Specifically, we fine-tuned the Mask R-CNN to segment the trachea and the ETT regions for 80 k iterations with the stochastic gradient descent (SGD) optimizer with momentum. The learning rate was set to 2.5 × 10^−4^, and we used a batch size of 128 for training.

#### 2.4.2. Network for Multi-Label Classification for Proper Positioning of the ETT in the Trachea

To detect proper positioning of the ETT in the trachea, we used the conventional EfficientNet B0 model for the multi-label classification problem. Unlike the original EfficientNet, which takes a colored image as an input, we utilized the segmentation results to elevate the classification performance. To be specific, we generated a 3-channel input including a gray-scale radiograph image and two segmentation results of trachea and tube from the segmentation network. In addition, we cropped the 3-channel image using the bounding boxes predicted by Mask R-CNN, and then resized the cropped image to 224 × 224 resolution, as shown in Figure 3. We used the initially pretrained EfficientNet on the ImageNet dataset and fine-tuned the network parameters with our own dataset. During training, we minimized the cross-entropy loss with the Adam optimizer [29]. The learning rate was fixed to 1 × 10^−5^ and the batch size was set to 32.

### 2.5. Experiments

We separately trained the proposed networks for the segmentation and classification tasks, and five-fold cross-validation was used for the performance evaluation with the training and validation set consisting of segmented images. Chest radiograph images were randomly divided into five parts; four out of the five images were used for training and the other was used for validation. Using this strategy, we could train the five models with five different training and validation datasets. Finally, we trained the optimized model with a total set consisting of segmented images and tested the performance of the ETT position classification using a test set consisting of images without segmentation mask.

### 2.6. Primary Outcomes

Our primary outcome was a favorable performance for automatic segmentation and four-label classification of the relationship between the trachea and the tip of the ETT. To evaluate the performance of the segmentation model, we used the Dice coefficient (Dice) to estimate the degree of overlap between the ground truth area and the predicted area, as follows:$$\mathrm{Dice}=\frac{2\mathrm{TP}}{2\mathrm{TP}+\mathrm{FP}+\mathrm{FN}}$$

To evaluate the performance of the classification model, we used the module with accuracy, precision, sensitivity, specificity, and F1-score in five-fold validation and a test. These values were analyzed according to each label performance. The prediction label was selected according to the class of which the prediction probability was highest among four classes. Accuracy is the fraction of correct predictions over total predictions. Precision is the positive predictive value that is the fraction of true positives (TP) among true and false positive (FP). Sensitivity (recall) is the fraction of TP among TP and false negative (FN). Specificity is the fraction of TN among TN and FP. F1-score is the harmonic mean of precision and recall as follows: (1)$$F1\;\mathrm{score}\;=2\times \frac{\mathrm{precision}\times \mathrm{recall}}{\mathrm{precision}+\mathrm{recall}}$$

### 2.7. Statistical Analysis

Data were compiled using a standard spreadsheet application (Excel 2016; Microsoft, Redmond, WA, USA) and analyzed using NCSS 12 (Statistical Software 2018, NCSS, LLC, Kaysville, UT, USA, ncss.com/software/ncss (accessed on 30 August 2021)). Kolmogorov–Smirnov tests were performed to demonstrate normal distribution of all datasets. We generated descriptive statistics and presented them as frequency and percentage for categorical data and as either median and interquartile range (IQR) (non-normal distribution) or mean and standard deviation (SD) (normal distribution) for continuous data. Student *t*-tests or Mann–Whitney tests were used to compare the characteristics of the segmented and non-segmented data. *p*-values < 0.05 were considered statistically significant.

## 3. Results

A total of 2092 images, including 932 images without the ETT and 1160 images with the ETT, were collected from 934 patients. These 934 patients consisted of 588 males (63%) with a mean [SD] age of 63.5 [18.3] years. A total of 107 images were excluded according to the exclusion criteria: 94 were images with tracheostomy tubes, and 13 images of poor quality by trauma (such as severe damage to anatomical structures or massive subcutaneous emphysema). Finally, a total of 1985 images, including 891 images without the ETT and 1094 images with the ETT were included in our study. Of the 1094 images with ETT, there were 129 images with shallow ETT position, 817 images with proper ETT position, and 148 images with deep ETT position. 

Out of the total number of images, 298 images without ETT, 97 images with shallow ETT position, 100 images with the ETT in the proper position, and 101 images with the ETT in deep position were segmented for training and five-fold validation. The distances between the carina of the trachea and the tip of the ETT were not significantly different between segmented and non-segmented datasets in all labels (all *p* > 0.05) (Table 1).

### 3.1. Five-Fold Validation for Automatic Segmentation of the Trachea and the ETT and Classification of Four Labels According to the ETT Position

The results for the five-fold validation of automatic segmentation and classification of four labels are shown in Table 2. Dice [mean (SD)] between segmented and predicted masks were 0.841 (0.063) for the trachea and 0.893 (0.078) for the ETT through five-fold validation. Values of the ETT according to position of the ETT were 0.879 (0.113) for shallow position, 0.908 (0.055) for proper position, and 0.892 (0.045) for deep position. Overall accuracy, F1 score [mean (SD)] of five-fold validation was 0.889 (0.034) (Table 2).

### 3.2. Performance Test of Each Label for Classification of Four Labels with Non-Segmented Images after Training with All Segmented Images

The confusion matrix and outcomes of the test for classification are shown in Table 3. Overall values of accuracy, precision, sensitivity, specificity, and F1-score were 0.922, 0.843, 0.843, 0.922, and 0.843, respectively. All values were highest in the label of absence among the four labels. Accuracy had the lowest value of 0.853 in the label of the proper position, and F1-score had the lowest value of 0.325 in the label of the shallow position. Sensitivity was higher in the shallow and deep positions than in the proper position, whereas precision values were the reverse.

## 4. Discussion

In the review article on computer-aided assessment of catheter and tube positions on radiographs, the authors recommended answering five-stage research questions covering presence, detection of the tip, course, type, and satisfactory position, as a system of evaluating catheter placement [30]. Compared to a general binary classifier, which may predict ETT absence/presence or ETT proper/improper position, our proposed method with automatic segmentation and classification of ETT and trachea using deep learning could provide four labels: absent, shallow, proper, and deep placement of ETT tip relative to the carina. 

In a study on segmentation of ETT and trachea images using deep learning and chest radiographs, the mean Dice scores of the ETT and the carina were 0.861 and 0.574, respectively, in the validation set [20]. The authors manually segmented the distal two-thirds of the ETT and carina, drawing a triangle that outlined the visible border of the main stem bronchi. In our study, the mean (SD) Dice for segmentation of the ETT and trachea were 0.841 (0.063) and 0.893 (0.078), respectively, in five-fold validation. We segmented the entire ETT and trachea shown in the chest radiograph. 

Previous studies have almost all used computerized methods to detect the presence of an ETT with results of approximately 0.93 (sensitivity) but did not assess the proper position of the ETT, which is a more crucial clinical judgment [16,31,32]. Recently, several studies have reported detection of the proper position including the presence of the ETT, or prediction of the distance from the tip of the ETT to the carina, using chest radiographs. Kara et al. reported that the performance of binary classification of presence or absence improved to 0.971 (accuracy), 0.971 (precision), and 0.974 (sensitivity) after identification and localization of the ETT using 16,000 images on the open-source MIMIC chest radiographs (MIMIC-CXR) dataset [19]. They also showed that the distance between the carina and distal ETT tip was predicted within a median error of 6 mm from manual ground-truth annotations in the overall final CNN assessment. Frid-Adar et al. reported that performance in detection of the ETT also reached an area under the curve (AUC) of 0.987, sensitivity of 0.955, and specificity of 0.965 with a test set of National Institutes of Health (NIH) dataset after training with several thousand synthetic and real images. However, these studies did not train the classification of proper ETT positions [32]. Our proposed method showed favorable performance for the detection of ETT absence among four labels, achieving 0.985 (accuracy), 0.986 (precision), 0.978 (sensitivity), and 0.990 (specificity) in the test set with non-segmented images after training with about 600 segmented images.

Lakhani trained a GoogLeNet convolutional neural network (CNN) to determine the presence/absence and low/normal position of the ETT on 300 labeled chest radiographs without automatic segmentation of the ETT and trachea, and achieved an AUC of 0.989 and 0.809, respectively [17]. In a study using approximately 23,000 chest radiographs and Inception V3 deep learning architecture, authors classified images into 12 categories, according to the distance from the carina and the tip of the ETT, at 1.0 cm intervals. The sensitivity and specificity for detecting shallow positioning of the ETT (i.e., distance > 70 mm) were 0.665 and 0.924; the sensitivity and specificity for detecting deep positioning of the ETT (i.e., distance < 20 mm) were 0.901 and 0.924 [18]. The authors found that their network misclassified the ground truth of 70–80 mm above the carina to a prediction of 60–70 mm, which negatively impacted sensitivity. We also achieved high-performance outcomes of 0.844 (sensitivity) and 0.921 (specificity) for detecting shallow positioning of the ETT (i.e., distance > 70 mm), and 0.894 (sensitivity) and 0.930 (specificity) for detecting deep positioning of the ETT (i.e., the distance < 30 mm), while the precision and F1-score were low, at about 0.3~0.5 for shallow and deep positioning of the ETT. In this study, we decided the proper depth of the ETT was in the range of 30 mm and 70 mm. The precision value might be lower than other values because the mean (SD) of the distance in the images was 78.56 mm (6.79 mm) in the shallow position and 19.09 mm (8.32 mm) in the deep position, which is near the proper depth of the ETT. Our proposed method could be helpful as a screening tool to detect excessively shallow and deep positions of the ETT, which could be critical for the patient.

There were several limitations in this study. First, the data of the chest radiographs and the patients originated from a single center, and our proposed model might not be suitable for the environment of other hospitals. Second, we decided that the distance for proper positioning of the ETT was 30–70 mm. This standard could be different according to other hospitals. Third, this method could not detect esophageal intubation, because there were no images of esophageal intubation in our dataset. Fourth, we did not assess how different lung conditions (such as pleural effusion, pneumonic infiltration, pneumothorax, and the underlying pathology of the respiratory insufficiency that led to intubation) influence the results of our proposed CNN method. Finally, we did not compare the performance of our model to that of physicians with respect to key factors such as clinical outcomes, the time required to reach a diagnosis, and the equipment required to use the model as a screening tool.

## 5. Conclusions

Automatic segmentation of ETT and trachea images and classification of the ETT position using CNN with plain chest radiographs could achieve good performance and improve the physician’s performance in deciding the appropriateness of the ETT depth.

## Figures and Tables

**Figure 1 jpm-12-01363-f001:**
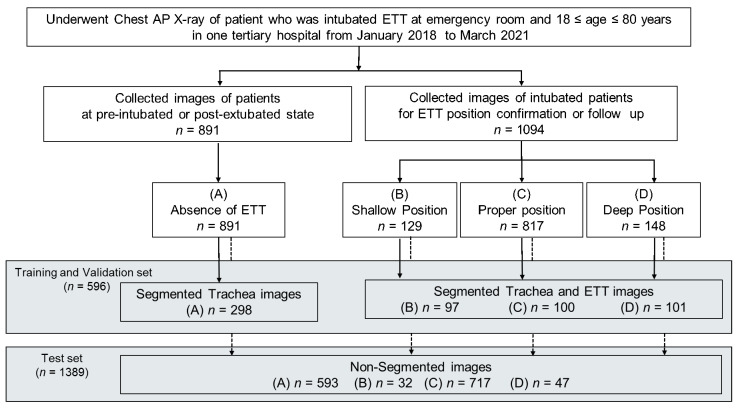
Flowchart of data collection for training/validation with segmented images and a test set with non-segmented images in the study.

**Figure 2 jpm-12-01363-f002:**
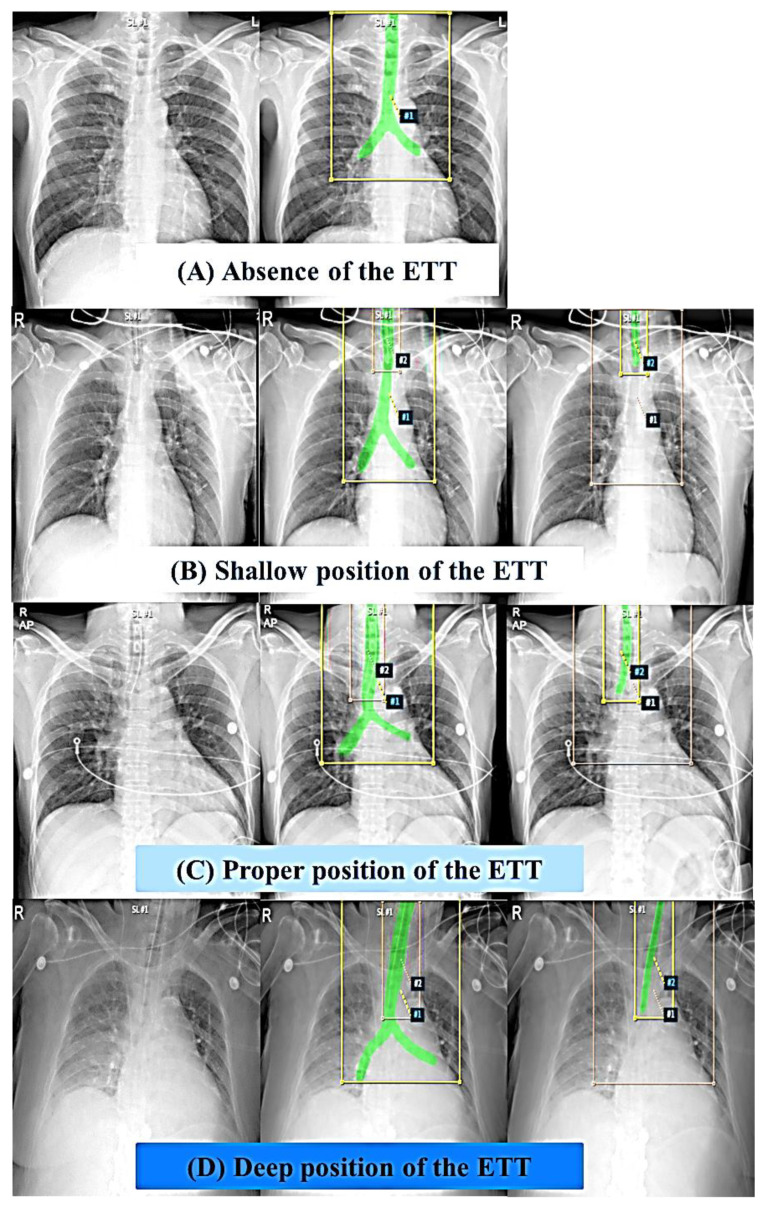
Segmentation of the trachea and the ETT on images according to four labels. (**A**) Absence of the ETT. (**B**) Shallow position of the ETT. (**C**) Proper position of the ETT. (**D**) Deep position of the ETT. Both the trachea and ETT were segmented on images with ETT, whereas only the trachea was segmented on images without ETT, and these segmented images were stored as Nifty (Neuroimaging Informatics Technology Initiative).

**Figure 3 jpm-12-01363-f003:**
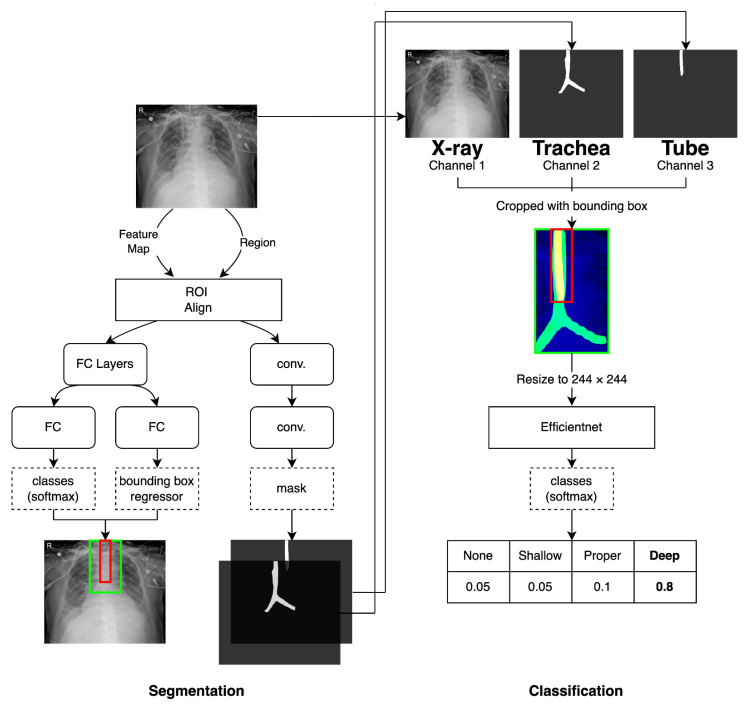
Overall flow of network architecture for segmentation of the trachea and ETT and for multi-label classification for proper position of the ETT in trachea. Mask R-CNN outputs segmentation results and bounding boxes. These results were fed into Efficientnet as an additional input to improve the multi-label classification accuracy. FC, fully connected layers.

**Table 1 jpm-12-01363-t001:** Comparison of the distances from the carina of the trachea to the tip of the ETT between segmented and non-segmented datasets according to the labels.

		The Distance from the Carina to Tip of the ETT, Mean (SD)		
		Segmented Data For the Training and 5-Fold Validation	Non-Segmented Data For the Test	*p*-Value
Labels	Absence	-	-	-
	Shallow, mm	78.76 (9.83)	78.56 (6.79)	0.915
	Proper, mm	48.68 (11.79)	47.60 (10.53)	0.343
	Deep, mm	20.56 (6.85)	19.09 (8.32)	0.261

ETT, endotracheal tube; SD, standard deviation.

**Table 2 jpm-12-01363-t002:** Outcomes of five-fold validation for automatic segmentation of the trachea and the ETT and classification of four labels according to the ETT position.

	1-Fold	2-Fold	3-Fold	4-Fold	5-Fold	Average of 5-Fold, Mean (SD)		
Segmentation								
Dice of the trachea, mean (SD)	0.840 (0.069)	0.843 (0.044)	0.847 (0.057)	0.846 (0.070)	0.832 (0.068)	0.841 (0.063)		
Dice of the ETT, mean (SD)	0.895 (0.065)	0.881 (0.132)	0.886 (0.060)	0.908 (0.041)	0.896 (0.056)	0.893 (0.078)		
						Shallow	Proper	Deep
						0.879 (0.113)	0.908 (0.055)	0.892 (0.045)
Classification								
Overall accuracy (F1 score)	0.942	0.867	0.882	0.899	0.856	0.889 (0.034)		

ETT, endotracheal tube; Dice, Dice coefficient; SD, standard deviation.

**Table 3 jpm-12-01363-t003:** Confusion matrix (A) and outcomes (B) of performance test for classification of four labels with non-segmented images after training with all segmented images.

**(A) Confusion Matrix**		**Prediction**				
		**Absence**	**Shallow**	**Proper**	**Deep**	**Sum**
Labels	Absence	580	5	4	4	593
	Shallow	3	27	2	0	32
	Proper	3	102	522	90	717
	Deep	2	0	3	42	47
	Sum	588	134	531	136	
**(B) Outcomes**		**Prediction**				
		**Absence**	**Shallow**	**Proper**	**Deep**	**Total (*n*)**
	True Positive, *n*	580	27	522	42	1171
	False Positive, *n*	8	107	9	94	218
	True Negative, *n*	788	1250	663	1248	3949
	False Negative, n	13	5	195	5	218
	Accuracy	0.985	0.919	0.853	0.929	0.922
	Precision	0.986	0.201	0.983	0.309	0.843
	Sensitivity	0.978	0.844	0.728	0.894	0.843
	Specificity	0.990	0.921	0.987	0.930	0.922
	F1 score	0.982	0.325	0.837	0.459	0.843

## Data Availability

The data presented in this study are available on request from the corresponding author.

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
