# Peer review of "Position Classification of the Endotracheal Tube with Automatic Segmentation of the Trachea and the Tube on Plain Chest Radiography Using Deep Convolutional Neural Network"

_jpm, 2022, doi:10.3390/jpm12091363_

Round 1
Reviewer 1 Report
In this article the authors present the convolutional neural network based algorithm to detect the placement of an endotracheal tube (ETT) in intubated patients from x-ray chest images. The novelty of this work is to detect not only proper/improper ETT placement but also shallow or deep ETT placement (improper placements). The topic of ETT placement in intubated patients is clinically relevant.
The article present well done work, however some explanations are necessary:
What were the criteria to divide the images into 1389 non-segmented and 596 segmented but not for example 50/50%? The segmented images with the absence of ETT (subgroup A) is two times more than in other subgroups (shallow, proper, deep – B, C, D). Why there is no about 100 in the subgroup A like in the others, for example?
The statistical results are only mean and standard deviation what means that only parametric tests by t-student were performed? If there is no results of non-parametric tests why Mann-Whitney test is mentioned in the methods section?
All requirements for parametric test were fulfilled? There is for example much more non-segmented images than segmented – compared groups (1389 vs 596).
Table 1 – the values of the (mean) measured distance from carina to the ETT tip are about 3 - 8 mm but in the text is mentioned: >70mm for shallow ETT placement, <30mm for deep ETT placement etc. It seems like these values in the table are “divided by 10” but It should be explained by authors, maybe I miss something.
Table 2 – how the classification accuracy were calculated? It is not F1-score?
Table 3 – The weakest results are Precision for “Shallow” and “Deep” (the most important cases from clinical point of view) due to the relatively high false positive values. It is because of small set of test data (32 and 47 images respectively)? Shallow and Deep placement were really improper ETT placement or “quite proper” but under/over the specified limit (70mm/30mm respectively)?
Table 3 – What could be a reason for relative high false negative value for “Proper” class.
Some minor remarks:
Please define the abbreviations in the abstract (e.g. ETT, CNN) when they are first used.
In the Paragraph 2.1 “This study was approved by the Institutional Review Boards of Hanyang ….” It should by “This study was approved by the Institutional Review Boards (IRB) of Hanyang….” As IRB abbreviation is used later
In the Paragraph 2.3 “An author (Jung HC)….” It should be: “The author (Jung HC)”
“Nifty (Neuroimaging informatics technology initiative). It is defined 2 times, the second one at the end of the Paragraph 2.3
In the Paragraph 2.4.1 at the end “bath size” should be “batch size”
Abbreviation NIH is not defined
AUC – the definition “area under curve” is present when used second time, not first time in the text
FC abbreviation in the Figure 3 is not explained in the text
Author Response
Thank you for your comprehensive analysis and study of our research. We agreed with your evaluation of our research and updated the paper to clarify each and every issue. We sought to increase the paper's depth in response to your remarks.

Reviewer 2 Report
I read with great interest the paper of et Jung H.C. et al. about automatic segmentation of ETT and trachea images and classification of the ETT position using CNN with plain chest radiographs. In congratulate the authors to their findings that could improve the physician’s performance in evaluating the proper position after endotracheal intubation, but would still ask for some minor revisions and answer to some questions:
page 2, please correct the proper verification of endotracheal tube placement:
· physical examination methods (auscultation of chest and epigastrium, visualization of thoracic movement, fogging in the tube), as well as pulse oximetry and chest radiography are not sufficiently reliable. Chest radiograph is not the gold standard for confirming the correct position of ETT, but certainly forms an important part of the overall assessment
· capnography can be used in patients who have adequate tissue perfusion
· ultrasound imaging may be used by someone who is experienced in this technique
· bronchoscopy is the gold standard
page 3, please complete the numbers: how many patients were initially included, how many patients were then excluded by which criteria (tracheostomy tubes, poor quality due to severe damage to anatomical structures or massive subcutaneous emphysema by trauma etc.).
Furthermore, I miss information about the underlying pathology of the respiratory insufficiency leading to the intubation or its correlate in the imaging.:
· What was the severity of the underlying pathology and its influence on the radiograph (RALE-Score, …)?
· How do pathologies such as pleural effusions, pneumonic infiltrates or a pneumothorax in general affect the result of the examination?
· How many of the investigated patients had enteric tubes and overlying catheters?
· Did devices with similar features as a linear appearance and brighter pixel intensities e.g., nasogastric tubes, interfere the assessment?
Author Response

(The authors gave the same response as above.)
